# A Simple Printed Cross-Dipole Antenna with Modified Feeding Structure and Dual-Layer Printed Reflector for Direction Finding Systems

**DOI:** 10.3390/s21175966

**Published:** 2021-09-06

**Authors:** Kyei Anim, Bonghyuk Park, Hui Dong Lee, Seunghyun Jang, Sunwoo Kong, Young-Bae Jung

**Affiliations:** 1Electronics Engineering Department, Hanbat National University, Daejeon 34158, Korea; kyeianim@gmail.com; 2Communication RF Research Section, Radio & Satellite Research Division, Electronics and Telecommunications Research Institute (ETRI), Daejeon 34129, Korea; bhpark@etri.re.kr (B.P.); leehd@etri.re.kr (H.D.L.); damduk@etri.re.kr (S.J.); swkong@etri.re.kr (S.K.)

**Keywords:** coaxial bead, cross-dipole antenna, direction finding, dual-layer reflector, right-hand circular polarization, through-hole signal via

## Abstract

In this paper, a simple printed cross-dipole (PCD) antenna to achieve a right-hand circular polarization (RHCP) at the L/S-band for direction finding (DF) systems is presented. The radiating part of the antenna consists of two printed dipoles that interlock with each other and are mounted orthogonally on a dual-layer printed reflector. To connect the feedlines of the dipole elements to the antenna’s feed network, which is located on the backside of the reflector, a through-hole signal via (THSV) is employed as the signal interconnection instead of the mainstream approach of using coaxial bead conductor. This feeding technique provides a degree of freedom to control the impedance of the signal path between the feedlines and the feed network in the numerical simulation for improved matching conditions. The proposed THSV extending through the dual-layer printed reflector is more reliable, durable, and mechanically robust to stabilize the matching conditions of the fabricated antenna in contrast to the coaxial-based approach that is more susceptible to impedance mismatch due to solder fatigue. Thus, the proposed PCD antenna offers advantages of broadband, flexible impedance matching, and fabrication ease. The antenna exhibits an impedance bandwidth (IBW) of 59% (1.59–2.93 GHz), a 3-dB axial ratio bandwidth (ARBW) of 57% (1.5–2.7 GHz), and a peak of 7.5 dB within the operating frequency band.

## 1. Introduction

Cross dipole antennas are mostly employed in various wireless applications such as direction finding (DF), where circular or dual-polarization is required to operate. Thus, several printed cross-dipole (PCD) antennas [1,2,3,4,5,6,7,8,9] have been previously reported in the literature to obtain circular polarization. In [8,9,10,11,12,13], the PCD antennas employ novel radiating structures or feeding techniques to achieve broadband performance and a wider axial ratio beamwidth (ARBW). A typical PCD antenna consists of two printed dipole elements, a metal reflector, and a feed network found at the backside of the reflector. In some cases, the integrated balun feedlines are incorporated into the design to excite the dipoles to enhance the impedance bandwidth (IBW) of the conventional PCD antenna to about 40–50%, as proposed by Li et al. in [14].

To connect the feedlines of the dipoles, which are mounted above the reflector, to the feed network of the PCD antenna located at the backside of the reflector, the mainstream approach is to use a commercial coaxial bead conductor extending through the reflector to route the signals. This design approach of the PCD antenna feeding scheme is cost-effective, simple to implement, and easy to fabricate. The microstrip-to-coax/coax-to-microstrip transitions are accomplished through soldering to commute the signals to and fro between the feedlines of the dipole elements and the feed network. However, the solder joints of the transitions often experience solder fatigue during the fabrication and assembling of the various parts of the PCD antenna. This occurrence limits the IBW of the fabricated model of the PCD antenna when compared to the numerical simulation results. Additionally, the coaxial bead conductor offers no degree of freedom to tune the impedance of the signal path between the dipole’s feedline and the feed network in the simulation model of the antenna for optimization purposes due to its fixed impedance of 50 Ω. Thus, the matching conditions of the PCD antenna in the numerical simulations may differ from that of the fabricated model to experience an IBW reduction. Although a good IBW is achieved with the coaxial bead in the simulation, there is no guarantee that such performance will be retained in the fabricated model, making this approach less reliable. Nonetheless, little to no attention has been given to the signal interconnection between the feedlines of the radiating dipoles and the feed network of the PCD antenna, which plays a critical role in stabilizing the matching conditions in the fabricated model.

The focus of this communication is, thus, to develop a PCD antenna at L/S-band whose signal interconnection is more reliable than the coaxial-based approach. The proposed PCD antenna utilizes a through-hole signal via (THSV) extending through a dual-layer printed reflector to interconnect between the feedlines of the dipoles and feed network of the antenna. With the THSV structure, the signals transition from the feed network to the feedlines and vice versa without relying on solder joints. Thus, the proposed PCD antenna with the THSV does not suffer from solder fatigue, making it more reliable than the conventionally used coaxial bead in the fabricated model. Additionally, the THSV structure offers flexibility in terms of freedom to tune the impedance of the signal path between the dipoles’ feedlines and the feed network in the numerical simulation. In this PCD design, the radiating structure constitutes two conventional printed dipoles and a feeding structure based on a THSV extending through a printed reflector to achieve an antenna with broadband, fabrication ease, mechanical robustness, low cost, and flexible impedance matching.

## 2. Antenna Design

The geometry of the proposed PCD antenna is illustrated in Figure 1. The antenna consists of a pair of printed dipoles (i.e., +45° and −45° linearly polarized dipoles) that interlock with each other, constituting an X-shaped radiating structure. The −45° dipole lies along x-axis while the +45° dipole lies on the *y*-axis to generate right-hand circular polarization (RHCP). The dipoles are fabricated on an RF-35 dielectric substrate (Taconic, Petersburgh, NY, USA; relative permittivity $\varepsilon_{r}$ = 3.55, loss tangent tanδ = 0.0025, and thickness $t_{sub}$ = 1 mm). Each dipole has a pair of top loads to make them achieve a more compact size. The dipoles are excited by the integrated balun feedlines etched on the front of the substrate to produce broadband performance. As seen in Figure 1, the radiating structure is mounted orthogonally above the dual-layer printed reflector by inserting the bottom end of the dipoles into the X-shaped assembly hole cut into the upper substrate. This makes it easy to assemble the antenna and further increases the antenna’s robustness when the dipole elements are eventually soldered to the reflector’s ground plane to create electrical paths for the return currents. The reflector component of the antenna consists of three copper layers, i.e., top layer M1, middle or isolating layer M2, and bottom layer M3. Metal ground vias are installed in its upper substrate around the signal vias to connect the top and middle ground planes (M1 and M2) to suppress leakage of the return currents from the ground plane to the signal via and vice versa. Also, additional ground vias are installed in the lower substrate to connect the middle and bottom ground planes (M2 and M3). The feed network of this PCD antenna is integrated on the lower substrate at the backside of the reflector so that the radiators and the feed network reside on separated substrates to achieve a feasible design and avoid short circuits between the dipoles and feed network circuit lines. Referring to Figure 1, clearances (air gaps) are introduced between the signal vias and the reflector’s ground planes to prevent short circuits.

The design parameters of the radiating dipoles are labeled out in Figure 2. The two dipoles are non-identical only in terms of feedline height $h_{f}$ to prevent feedline overlap as the dipoles interlock and dipole length $\;l_{d}$ to improve the axial ratio of the antenna. Because the two dipoles have somewhat different dimensions, their input impedances slightly differ at various frequency points within the operating bandwidth, as illustrated in Figure 3. It shows that the +45° and −45° dipoles assume input impedance of (47.6 +j8.41) Ω and (56.7 +j7.75) Ω, respectively. It should be noted that the real impedances Re {Z} are low and close to 50 Ω to match the 50 Ω SMP connector.

The reflector component of the proposed PCD antenna has a dual-layer printed topology (see Figure 1). Figure 4 shows the top and back views of the reflector used in the antenna design. It is fabricated on FR-4 dielectric substrate ($\varepsilon_{r}$ = 4.3, tanδ = 0.025, thickness $t_{1}$ = 1 mm, $t_{2}$ = 1.5 mm), which was chosen due to its robustness without requiring metal support. As seen in Figure 4a, the X-shaped hole milled into the upper substrate of the reflector is used to hold in place the radiators on the reflector. Integrated on the substrate are metal pads functioning as the electrical contacts to directly connect the feedlines of the dipoles to the signal vias in reflector’s substrates (see Figure 4a). Following the signal vias are the lower metal pads that join the signal vias to the feed network located at the backside of the reflector, as demonstrated in Figure 4b, forming the antenna’s feeding structure. The configuration of the feed network is shown in Figure 4b. It is composed of two meander lines, a 3 dB-hybrid coupler, and two branched lines (one leading to the SMP connector as input and the other 50 Ω-isolation). The hybrid coupler is a commercial X3C17A1-03WA (Anaren, NY, USA) that operates at 690–2700 MHz with isolation of 20 dB_min_, voltage standing wave ratio (VSWR) of 1:33, impedance of 50 Ω and insertion loss of 0.49 dB_max_. The meander lines allow for a time delay to compensate for the small misalignment between the signal vias of the two dipoles in the vertical axis. The feed network, thus, generates an equal magnitude and a 90° phase difference between the two dipoles for an RHCP. The dimensions of the proposed antenna in Figure 1 obtained through electromagnetic optimization are detailed as follows (unit: mm): $\;g_{1}=3.44,\;h_{a}=6.25,\;h_{b}=8.57,\;h_{f1}=20.27,\;h_{f2}=17.47,\;h_{s}=48.48,\;h_{p}=33,\;\;h_{t}=10,\;l_{d1}=36,\;l_{d2}=34,\;l_{f}=15.93,\;w_{f}=2.63,\;\mathrm{and}\;s=0.5$.

### 2.1. Feeding Structure Based on Coaxial Bead Conductor

As mentioned earlier, the conventional approach to route the signals up and down between feedlines of the dipoles and the feed network is through a coaxial bead conductor, as shown In Figure 5a. It is easily implementable and cost-effective. However, as seen in Figure 5a, the microstrip-to-coax/coax-to-microstrip transitions can only be accomplished through soldering. Thus, the fabricated model of the PCD antenna with the coaxial bead is susceptible to solder fatigue during fabrication and assembly, which eventually leads to impedance mismatch. This phenomenon is more severe at the perpendicular microstrip-to-coax, as shown in the conventional case in Figure 5a. Also, the coaxial bead offers no flexibility to control the impedance of the signal path between feedlines and the feed network in the numerical simulation due to the fixed impedance of 50 Ω. Hence, the IBW obtained in the numerical simulation could be limited by the fabricated model owing to different matching conditions. It can be seen from Figure 5a that the coaxial bead extends through an aluminum reflector, which is the traditional design approach, to cause the return signals to traverse to and fro between the coaxial outer conductor and the reflector. The coaxial bead and the metal reflector must be properly and tightly soldered to avoid friction to cause reflections to limit bandwidth in the fabricated antenna. The combination of the above factors, thus, makes the coaxial-based approach less reliable and less robust to maintain the performances of the simulated model during the fabrication and assembling of the antenna.

### 2.2. Proposed Through-Hole Signal via (THSV) Design

Here, a through-hole signal via (THSV) structure is employed as a practical solution to the issues related to the coaxial-bead approach, as illustrated in Figure 5b. It offers however the same benefits as the conventional case, such as fabrication ease and low cost. The feeding structure consists of the feed network, the THSV structures extending through the dual-layer printed reflector, and the balun feedlines, which requires no solder joints to routes signals to the radiators. Hence, the feed network-to-signal via and signal via-to-feedline transitions are accomplished through metal pads, as demonstrated in Figure 5b. The proposed PCD antenna experiences no solder fatigue to provide reliability and robustness in the fabricated prototype.

The reflection coefficients of the PCD antennas (see Figure 5) are plotted in Figure 6a, which shows the measured and simulated $S_{11}$ of the proposed case, as well as the conventional case for the sake of fair comparison. Figure 6 shows that the IBW for $S_{11}<-$10 dB of the proposed PCD antenna with the THSV is about twice that of the conventional case with a coaxial bead. In the conventional case, compared to the measured $\;S_{11}$, the IBW of the antenna in the numerical simulation completely outperforms the measured result, which can be attributed to the factors mentioned early on. Meanwhile, in the proposed case, the measured $S_{11}$ is in good agreement with the simulated one, as shown in Figure 6a. Based on the reflection coefficients shown in Figure 6a, it can be conjectured that the signal interconnections between the feed network and the feedlines of the radiating dipole play a crucial role in improving and stabilizing the matching conditions of the antenna to achieve better performance. Figure 6b demonstrates the influence of the reflector size (length $\;G_{l}$ and width $\;G_{w}$) on the antenna gain and reflection characteristics ($S_{11}$). It could be observed that adjusting the size of the reflector has minimal impact on the gain and $S_{11}$ as it slightly improves the performances when the dimensions increase from 50 mm to 100 mm, beyond which the performances remain fairly constant. The optimal size of the reflection is fixed as $G_{l}$=$G_{w}$= 150 mm.

The full 3D view and the equivalent circuit model of the THSV structure are shown in Figure 7. As seen in Figure 7a, the bottom end of the feedline sits directly on the circular metal pad attached to the signal via that extends through the two substrates of the antenna’s reflector. The signal via is followed by the lower metal pad, which is integrated on the bottom substrate at the backside, to directly connect to the feed network microstrip line without soldering. The radii of the upper $r_{u}$ and lower $r_{l}$ metal pads for the two dipole elements were optimized and fixed as $r_{u1}$ = $r_{u2}$ = 0.6 mm, $r_{l1}$ = 2 mm, and $r_{l2}$ = 2.5 mm, respectively. To guarantee that no air gap exists between the feedline and the metal pad, a small solder is applied on the top surface of the upper metal pad to make it airtight. Additionally, the feedlines are held in place above the metal pads as the radiating structure is partially inserted into the X-shaped hole milled into the upper substrate of the reflector (see Figure 1). Thus, this feeding scheme is much more robust than the coaxial-based technique. 

However, the design of the THSV generates fringing capacitance $C_{via-GND}$ between the signal via and the ground planes of the reflector, as depicted in the circuit model in Figure 7b. As a result, the return currents leak to and fro between the signal via and the ground planes (see Figure 7a) through the $C_{via-GND}$ to cause strong reflected signals that limit the antenna’s total bandwidth. Meanwhile, the equivalent model of the conventional case in Figure 5a is demonstrated in Figure 7c.

To overcome this issue, metal ground vias are installed around the THSV structure to connect the top (M1) and middle (M2) ground planes in the upper substrate of the reflector (See Figure 1) to act as a continuous waveguide. Thus, the fringing capacitance $C_{via-GND}$ is suppressed, and the forward signals in the THSV propagate at the TEM mode characterized by minimum reflections to improve the matching conditions. The influence of the ground vias on the reflection coefficient $S_{11}$ has been demonstrated in Figure 7d.

As mentioned earlier in the previous section, the THSV offers the advantage of providing stable matching conditions in the proposed antenna, which has been demonstrated in Figure 8. It can be observed that the linear variation of the signal via size ($\;r_{v}$) is less sensitive to the impedance of the signal path between the feedline and feed network over wider span of frequencies, especially in the L and S-bands. Thus, once the radius $r_{v}$ of the THSV is fine-tuned to achieve an impedance of about 50 Ω to match with feedline and feed network microstrip lines, the performances become very stable and insensitive to impedance variation due to stable matching conditions. The radii of the THVS for the +45 and −45 dipole elements have been optimized and fixed as $r_{v1}\;$ = 0.12 mm and $r_{v2}$ = 0.22 mm, respectively, to obtain impedance of around 50 Ω within the band of interest (1.5–3 GHz), as illustrated in Figure 8. However, it should be noted that the THSV is very sensitive to impedance variation in the higher frequencies to yield stable matching conditions. Thus, the THSV will not operate optimally in the higher frequencies.

Finally, the proposed PCD antenna constitutes two printed dipoles (whose design is based on conventional method), a dual-layer printed reflector, and a feed mechanism based on the THSV and integrated balun feedline. It, therefore, offers promising features such as broadband, good axial ratio, impedance matching ease, geometrically simple, robustness, low cost, and fabrication ease.

## 3. Results and Discussions

Photographs of the fabricated antenna are shown in Figure 9. Its dimensions are 150 mm × 150 mm × 48.4 mm (L × W × H). The dual-layer printed reflector of the antenna has an X-shaped assembly hole in its upper substrate to firmly mount the radiating element above the reflector (see Figure 9b). The assembly hole also helps to hold in place feedlines of the dipole elements sitting on top of the upper-layer metal pads. The presence of the soldering pads in Figure 9b makes it possible to solder the radiator to the reflector. The feed network, integrated on the backside of the lower substrate of the reflector, as shown in Figure 9c, is geometrically simple to achieve RHCP.

The simulated and measured reflection coefficients, $S_{11}$ and the port-to-port isolation are plotted in Figure 10a. The measured $S_{11}$ shows an impedance bandwidth (IBW) of 59.29% from 1.59 to 2.93 GHz for $S_{11}<-$10 dB. The port-to-port isolation of the proposed antenna is greater than 35 dB within the band of interest. The measured 3-dB axial ratio bandwidths (ARBW) are shown in Figure 10b is 57.1% from 1.5 to 2.7 GHz, which ironically outperforms the simulated one that covers 1.8–2.58 GHz (35.6%). The measured gain of the proposed antenna is 4.19–7.5 dB within the operating band. The measured and simulated gain disparity is attributable to factors such as minor fabrication and measurement errors.

Figure 11a–c shows the simulated and measured E- and H-field radiation patterns at 1.7, 1.9, and 2.5 GHz. The antenna achieves a very good radiation pattern in the broadside with half-power beamwidths (HPBWs) of 65.9°, 69°, and 78.3° at these frequency points. The left-hand circularly polarized (LHCP) levels at the interest frequencies are lower than −6 dB in both the $E\;\left(xz\right)$- and $\;H\left(yz\right)$-planes, as shown in Figure 11.

Table 1 compares the proposed antenna to previously reported PCD antennas in terms of impedance bandwidth (IBW), 3-dB axial ratio bandwidth (ARBW), and peak gain. Unlike the other antennas in the table that employ complex dipole geometry and (or) sophisticated feeding structure to obtain broadband, the proposed antenna uses conventional dipoles coupled with a simple feeding scheme to achieve wideband. Thus, the proposed antenna has IBW lower than [10,11,13], which tend to have complex geometry regarding radiating or feeding structure. Although the IBW of the proposed antenna is lower than most antennas in Table 1, its ARBW is better than most of them apart from [10,13] with comparable gain. Also, apart from [6,10], the beamwidth of the proposed antenna is comparable to the other cross-dipoles in literature.

## 4. Conclusions

In summary, a printed cross-dipole antenna (PCD) at L/S-bands to obtain right-hand circular polarization has been designed. The antenna proposes a through-hole signal via (THSV) that extends through a dual-layer printed reflector interconnect between the feedlines of the dipole elements and the feeding network. Thus, the transition of the signals from the feed network to the feedlines and vice versa is accomplished without solder joints as opposed to the mainstream approach of using a coaxial bead conductor, which is susceptible to solder fatigue to cause impedance mismatch during the antenna fabrication. The proposed antenna constitutes two simple dipoles, a dual-layer printed reflector, and a feed mechanism based on the THSV and integrated balun feedline. It offers significant advantages such as broadband, good axial ratio, impedance matching ease, geometrically simple, robustness, low cost, and fabrication ease.

## Figures and Tables

**Figure 1 sensors-21-05966-f001:**
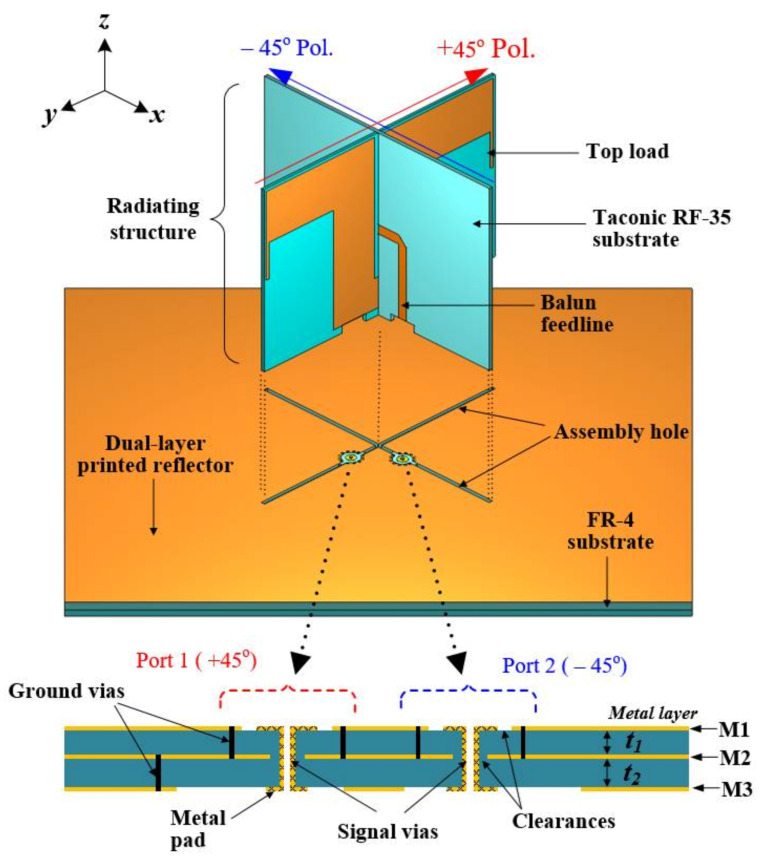
Configuration of the proposed antenna with printed dual-layer reflector.

**Figure 2 sensors-21-05966-f002:**
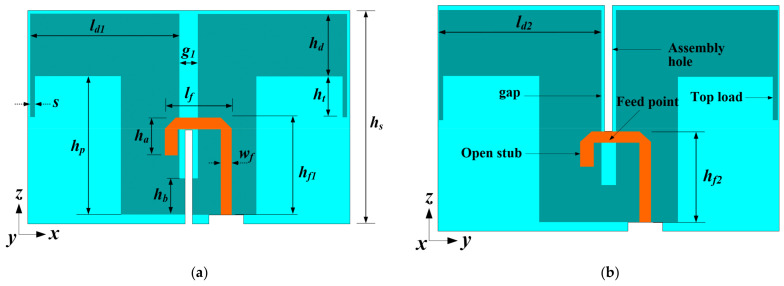
Geometry of the radiating elements with integrated balun feedlines; (**a**) +45-deg. (Port 1) and (**b**) −45-deg. (Port 2) linearly polarized dipoles.

**Figure 3 sensors-21-05966-f003:**
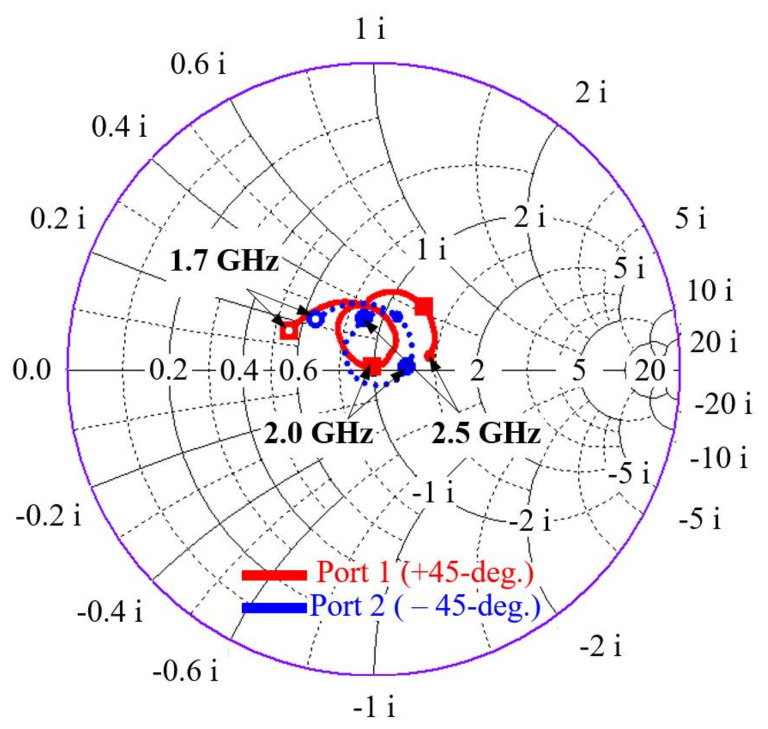
Simulated input impedance curves at different frequency points of the two dipoles.

**Figure 4 sensors-21-05966-f004:**
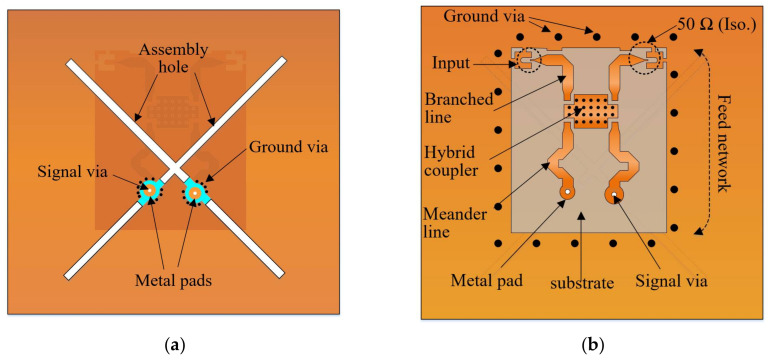
Geometry of the dual-layer printed reflector showing the feed network; (**a**) Top view; (**b**) Back view.

**Figure 5 sensors-21-05966-f005:**
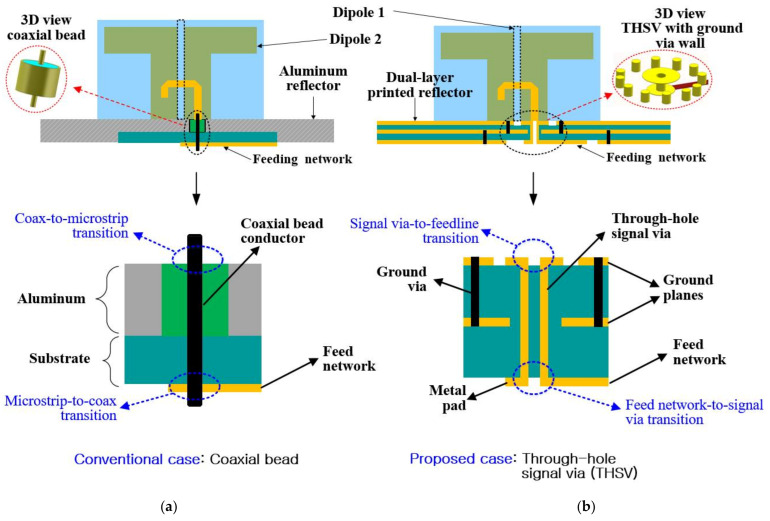
The feeding scheme of the printed cross dipole antenna showing: (**a**) Coaxial bead interconnection; (**b**) Proposed through-hole signal via (THSV) interconnection.

**Figure 6 sensors-21-05966-f006:**
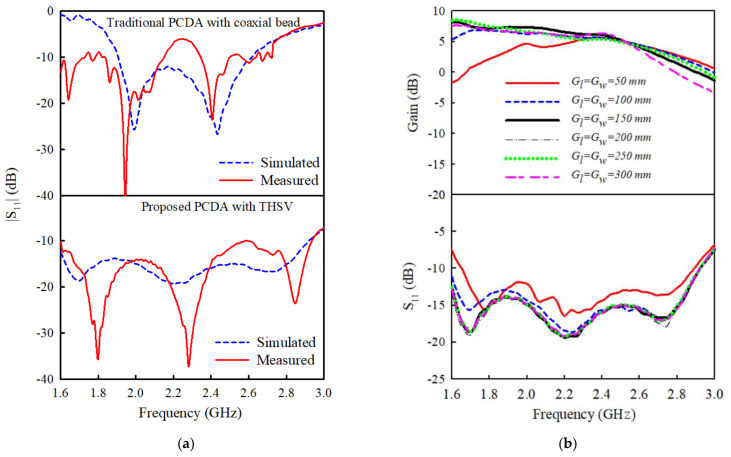
(**a**) Simulated and measured |$S_{11}$| of the PCD antenna with the proposed through-hole signal via (THSV) versus the conventional coaxial bead interconnection; (**b**) Variation of antenna gain and $S_{11}$ with reflector ground size (*G_l_*, *G_w_*).

**Figure 7 sensors-21-05966-f007:**
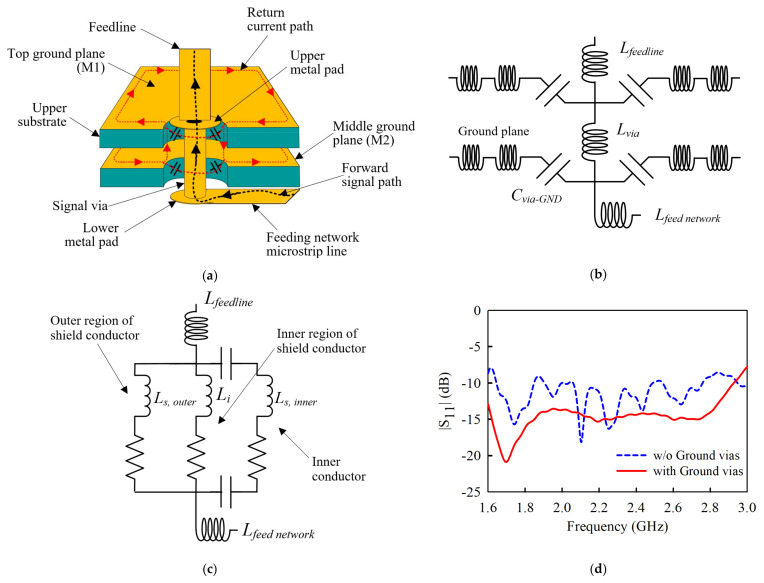
Proposed through-hole signal via (THSV) interconnection for the PCD antenna feeding scheme: (**a**) 3D model; (**b**) Circuit model of proposed case with THSV; (**c**) Circuit model of conventional case with coaxial bead; (**d**) The influence of the ground vias on the antenna’s reflection coefficient.

**Figure 8 sensors-21-05966-f008:**
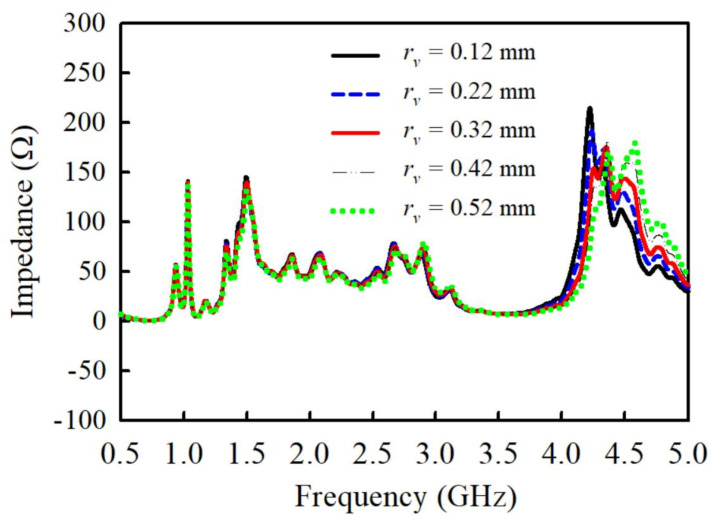
Simulated and measured |$S_{11}$| of the PCD antenna with the proposed through-hole signal via (THSV) versus the conventional coaxial bead interconnection.

**Figure 9 sensors-21-05966-f009:**
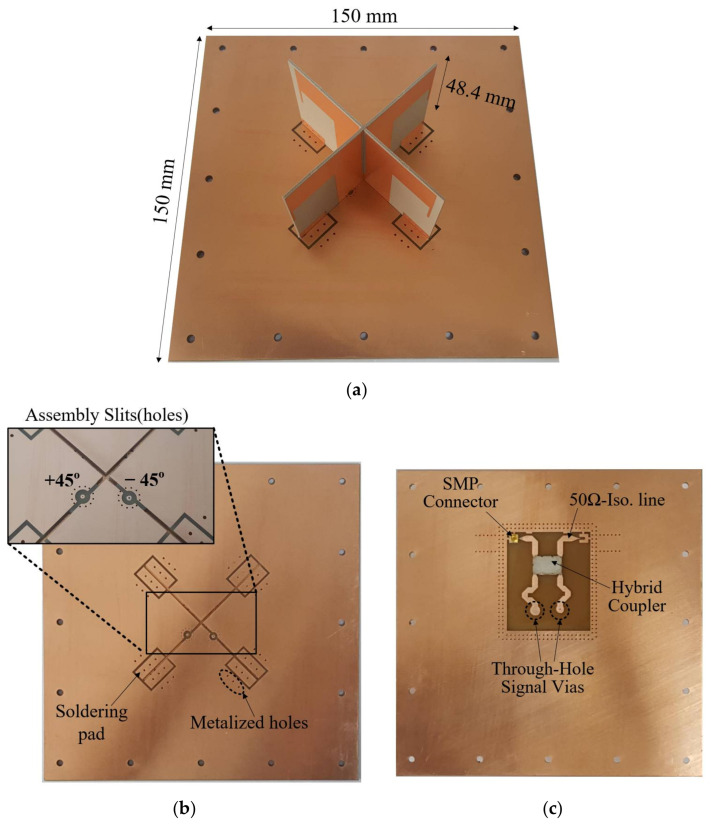
Photos of the fabricated antenna; (**a**) Perspective view; (**b**) Top view of the printed reflector showing the X-shaped assembly hole; (**c**) Back view of the reflector with the feed network configuration.

**Figure 10 sensors-21-05966-f010:**
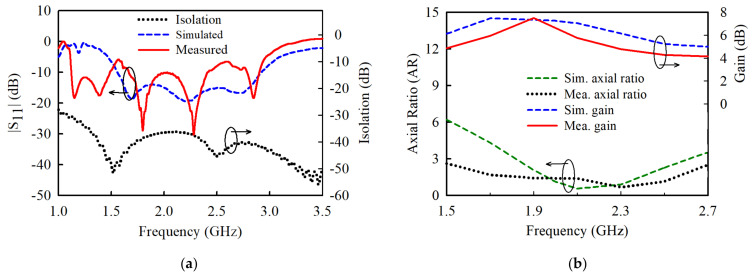
Experimental results of the proposed PCD antenna: (**a**) Reflection coefficients and port-to-port isolation. (**b**) Axial ratio and gain.

**Figure 11 sensors-21-05966-f011:**
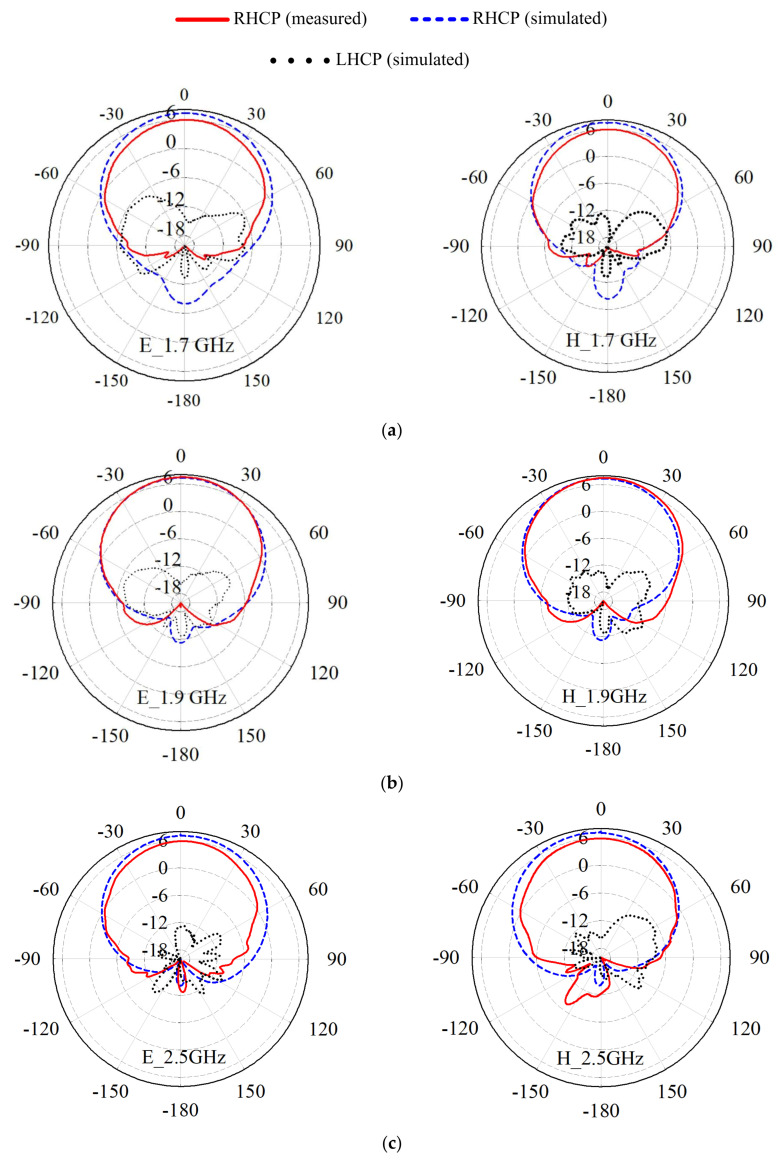
Simulated and measured E- and H-field radiation patterns of the proposed PCD antenna at (**a**) 1.7 GHz; (**b**) 1.9 GHz; (**c**) 2.5 GHz.

**Table 1 sensors-21-05966-t001:** Comparison of the existing printed cross-dipole and proposed antennas operating at L-/S-bands.

Antenna	*f* (GHz)	IBW (%)	ARBW (%)	Beamwidth (°)	Max. gain (dB)	Size (L × W × H)
Proposed	2.0	59.3	57.1	65.9–78.3	7.5	1λ × 1λ × 0.32λ
[2]	1.68	38.16	12.9	>60	8	H = 0.37λ (Aperture diameter = 1.086λ)
[6]	1.67	40	12.9	100–110	9	H = 0.456λ (Aperture diameter = 0.52λ)
[10]	2.89	70.6	62.4	94–95	6	0.674λ × 0.674λ × 0.37λ
[11]	0.84	79	36	-	8	0.98λ × 0.98λ × 0.33λ
[13]	1.57	66.7	57.7	-	0.9	0.52λ × 0.52λ × 0.65λ
[15]	2.04	57	39	-	10.7	H = 0.257λ (Aperture diameter = 1.028λ)
[16]	2.33	57.5	Dual Pol.	58.1–72.6 (E) 76.8–85.5 (H)	8.8	2.33λ × 1.126λ × 0.29λ
[17]	1.94	23.7	Dual Pol.	76.8–85.5	6.9	0.97λ × 0.97λ × 0.226λ

f is the center frequency and λ is free space wavelength of the operating band.

## Data Availability

Not applicable.

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
