# Peer review of "A Simple Printed Cross-Dipole Antenna with Modified Feeding Structure and Dual-Layer Printed Reflector for Direction Finding Systems"

_sensors, 2021, doi:10.3390/s21175966_

Round 1

Reviewer 1 Report

This manuscript proposed a wideband circularly polarized antenna based on cross-dipole. By using the modified feeding structure and the dual-layer printed reflector, more design parameters can be used to turn the impedance of the antenna, and extend the bandwidth. In general, it is an interesting idea, but the performances of the antenna only show limited benefit. Thus, I think the paper can be revised.

  1. The equivalent circuit of the conventional case in Fig. 5 can be added in Fig. 7, to give a more clear explanation of the proposed dual-layered connection.
  2. The influence of the ground size on the antenna performance should be discussed.
  3. Please mention that how to obtain the final dimensions of the proposed antenna. Are they obtained by optimization, or the circuit model?
  4. The details of the 3dB-hybrid coupler are missing.
  5. Since the AR bandwidth is mainly limited by the 3dB-hybrid coupler, which is not the core technology of the proposed design, I think that some other crossed dipoles in literature can be included in Table I, which can only provide dual-linear polarizations.
  6. Table 1 is desired to include the size of the antennas, and other important performances, e.g. the beamwidth, efficiency and etc.

Author Response

Please refer an attached file

Reviewer 2 Report

This communication is mainly about the design and efficient solutions of PCB antenna and its feed networks. It brings a new atypical solution in connection with this type of antenna. It replaces the conventional coaxial-based approach with a new PCB solution.

Regarding the methodology, author should quantitatively state the influence of the choice of vias parameters and impedance parameters.

In my opinion, there is a mistake in line 45 (... accomplished through soldering to commute the signals to and "fro"). Correct "from" .

Also in line 63 the "feed network" should be correct instead of "feed work".

"In this paper ..." is more often used instead of "In this communication ...". 

Author Response

Please refer an attached file

Reviewer 3 Report

1.Does the small solder in line 197 have any effect on the results?

2.Please analyze the effect of different THSV parameters on the final performance to further clarify the optimization process?

Author Response

Please refer an attached file

Round 2

Reviewer 1 Report

The authors have addressed all my comments in the revised manuscript. I have no more questions about the paper.